



# Wet season rainfall characteristics and temporal changes for Cape Town South Africa, 1841-2018

Nothabo E. Ndebele[1], Stefan Grab[2], Herbert Hove[1]

[1]School of Statistics and Actuarial Science, University of the Witwatersrand, South Africa

[2]School of Geography, Archaeology and Environmental Studies, University of the Witwatersrand, South Africa

*Correspondence to*: Nothabo E. Ndebele (Nothabo.Ndebele@wits.ac.za)

**Abstract.** Wet seasons may be characterized by the frequency of wet/dry days, duration of wet/dry spells, and season length. These properties are investigated for Cape Town using rainfall data from four weather stations in the Cape Town metropolitan area located at the South African Astronomical Observatory (SAAO), Maitland, Kirstenbosch and Cape Town

International airport. The primary focus is on the long SAAO daily rainfall record dating back to 1841, with the specific aim to statistically assess attributes of the wet season (April to October) and its temporal variability over the period 1841-2018. The decade 1950-1959 had significantly high frequencies of wet days, but there was a subsequent significant decline in wet days at the SAAO (-1day/decade) and Maitland (-1.1days/decade) during the period 1950-2018. A significant decline in wet days also occurred at the SAAO between 1880 and 1940 (-3.3 days/decade, $p = 0.005$). Dry spells longer than 5 days have

become more prevalent since the beginning of the 20[th] century. A rain-based definition for the onset and termination of the wet season is presented using 5-day running sums and pentad means; these were applied to each year containing adequate daily data, so as to track changes during the wet season. Mean season length over recent decades (1950-2018) is 186 days, but this has declined over this period of time such that it averages 183 days for the most recent *c.* 4 decades (1979- 2018). This decline is attributed to an increased incidence of late onsets (after 15 April) and early terminations (earlier than 18

October) of the wet season, or a combination of both, particularly since the year 2000. Interannual variability in wet season characteristics is associated with solar (sunspot) cycles and fluctuations in the Southern Oscillation Index and Southern Annular Mode.

## 1. Introduction

Temporal characteristics of seasonal rainfall vary across regions and climates of the world (Knoben et al., 2019). In Africa,

wet seasons are either unimodal (concentrated over only one peak rainfall period per 12 month cycle: e.g. southern Africa) or bimodal (two temporally separated wet seasons per 12 month cycle: e.g. south coast of South Africa and large parts of East and West Africa) (Camberlin et al., 2009; Amekudzi et al., 2015). Understanding the nature and potential changes to such wet seasons is important to several sectors, most particularly agriculture, water resources planning and health (Hachigonta et al., 2008; Singh and Ranade, 2010). To this end, considerable scientific interest has focused on establishing the



onset/termination dates of the annual wet season(s) and how these may have changed over recent decades (e.g. Mupangwa et al., 2011; Owusu and Waylen, 2013; Oguntunde et al., 2014; Obarein and Amanambu, 2019; Giráldez et al., 2020), or might change in future through climate modeling projections (e.g. Kusunoki et al., 2011; Sarr, 2012; Saeed et al., 2018). Defining seasonal rainfall onset/termination dates depends on location, climate and the study purpose; hence defining attributes have been variable across disciplines (climatology, hydrology, agronomy) and regions. Rainfall based variables are most

commonly used to determine rainfall onset/termination dates; these include the number of wet days, rainfall amounts (mm) and wet/dry spells (days)(e.g. Hachigonta et al., 2008; Ngetich et al., 2014). However, in some instances, other variables such as temperature, wind and humidity have also been used to compute potential evapotranspiration and soil water balance, particularly to help determine suitable rainfall thresholds required for specific crop growth (Mugalavai et al., 2008; Moron and Robertson, 2014).

Establishing wet season length usually requires rainfall based empirical methods. For instance, Boyard-Micheau et al. (2013) determine the rainfall onset date in Kenya and northern Tanzania based on the first wet day of $N$ consecutive days, receiving a minimum of $P$mm of rain, without a dry spell lasting more than $N_d$ number of days, and additionally specify a minimum $P_{min}$ value for subsequent days. Each of these parameter values is determined empirically to address a set of requirements, most typically associated with crop growth. So for example, understanding the rainfall requirements for maize to germinate

and grow has helped establish an appropriate onset date, based on a threshold of 25mm, but over a 10 day period followed by at least 20mm over the subsequent 20 days (Tadross et al., 2005; Hachigonta et al., 2008). In this case, rainfall data were divided into pentads (5 day periods) so as to identify the pentad that best satisfied the criteria for maize growth (Tadross et al., 2005). The use of pentads is also common practice for identifying other characteristics of the wet season, such as wet season length (e.g. Hachigonta et al., 2008; Marengo et al., 2017; Byakatonda et al., 2018).

Alternative approaches to establish seasonal rainfall onset/termination dates include using the cumulative distribution of rainfall amounts (Nicholls, 1984; Amekudzi et al., 2015) or accumulated rainfall anomalies (Liebmann and Marengo, 2001; Camberlin et al., 2009; Liebmann et al., 2012; Dunning et al., 2016). Liebmann and Marengo (2001) delineate onset/termination dates by obtaining absolute minimum/maximum of annual accumulated anomalies, calculated by subtracting the long-term mean rainfall from the actual rainfall for each day of the year. This approach is less sensitive to

those using specified thresholds and may thus be applied at local and regional scales. The method by Liebmann et al. (2012) has more recently been extended to accommodate regions that have two wet seasons per annum, such as parts of West and East Africa (e.g. Dunning et al., 2016).

Also of importance, is establishing rainfall characteristics (daily frequency, distribution, amount; wet and dry spells) during the wet season, as these may have an influence on crop production, biosystems, hydrology etc. Wet/dry days are defined by

either using a fixed value or using a parameter such as mean daily rainfall, for the area of interest. Given that the usual precision of rain gauges is 0.1mm, at times this is taken to be the threshold value (e.g. Martin-Vide and Gomez, 1999). However, many prefer a 1mm threshold to define a wet day given the assumption that an amount less than 1mm is likely returned to the atmosphere through evapotranspiration (e.g. Kruger, 2006; Nastos and Zerefos, 2009; Froidurot and





Diedhiou, 2017; Valdes-Abellan et al., 2017; Byakatonda et al., 2019). Most recently, Rivoire et al. (2019) argued that since
evapotranspiration is a variable (depending on the season), a fixed threshold may not be suitable, particularly for the dry
season.   When a wet day threshold has been defined, wet/dry spells are identified as consecutive days of wet/dry days, and
these may be used to further investigate rainfall characteristics.

Several previous studies have examined winter rainfall and attributes of the wet season over southernmost Africa (e.g.
Reason et al., 2002; Reason and Rouault, 2005; Dieppois et al., 2016; du Plessis, 2017; Mahlalela et al., 2019). Identified
associations between rainfall and climate modes such as the El Nino Southern Oscillation (ENSO), Southern Annular Mode
(SAM) and Sea Surface Temperatures (SST) are now well established for the region (Tyson, 1981; Mason and Tyson, 1992;
Reason and Rouault, 2002; Reason and Jagadheesha, 2005;  Philippon et al., 2012). The association of solar activity with
rainfall and other climate variables has also been confirmed for the region (Dyer, 1975; Mason and Tyson, 1992; Ndebele et
al., 2020) and other parts of  the world (e.g. Laurenz et al., 2019; Nitka and Burnecki, 2019). Despite such previous scientific
effort, to date, and to our knowledge, relatively few studies have investigated long-term (> a century) wet season
characteristics in Africa, despite its potential importance for ongoing and future agricultural planning at sub-regional scales.
To this end, we use one of the worlds' longest single station rainfall records (178 years), namely that of the South African
Astronomical Observatory (SAAO) in Cape Town, South Africa, to statistically establish the nature and temporal changes of
wet season characteristics through the period 1841-2018. The work expands on that recently describing the trends and
variability of rainfall in Cape Town since 1841 (Ndebele et al., 2020). While the SAAO record is the primary focus of this
paper given its long record and historic value, additional station records over the Cape Town metropolitan region are also
analyzed for imputation and comparative purposes. Specific objectives of the current paper include the quantification of wet
season length, the occurrence of dry/wet spells, frequency of wet days, and to establish the temporal variation of these
attributes. Such rainfall characteristics for Cape Town, which previous studies have not yet examined, provide important
long-term perspectives relevant to, in particular, agricultural production, hydrology, and ecosystem functioning. In addition,
given the considerable length of our dataset, we statistically test for any associations between these wet season rainfall
characteristics and their potential drivers (i.e. El Nino Southern Oscillation, Southern Annular Mode, and solar activity
[sunspots]).

## 90    2.   Site context and data

### 2.1.  Site context

Cape Town is the second largest city in South Africa and has a coastal setting in the southwestern portion of the Western
Cape Province (Fig. 1). This region is characterised by a complex topography with mountain ranges exceeding 1000m in
elevation within a few km of the coast, but also includes the Cape flats, a low lying and flat relief bounded by the Atlantic
Ocean on southern and western sides, and where Cape Town International Airport is located (Goodness and Anderson,





2013). The eastern side of the Cape Flats and False Bay also have mountain ranges which exceed 1500m in some places. Cape Town has a Mediterranean climate with typically cool, wet austral winters (May-Sept) and dry, warm summers (Ndebele et al., 2020). According to Mahlalela et al. (2019), the southwestern and eastern coastal regions, as also some windward facing mountain slopes, receive on average >500mm of rainfall per annum, yet parts of the SW Cape region,

particularly northwards and inland of Cape Town, may receive as little as 200-400mm per annum. Rainfall quantity/distribution is thus highly variable and influenced by both topography and direction of rain-bearing weather systems. Rainfall over the southwestern Cape region is primarily associated with overpassing cold fronts connected to mid-latitude cyclones, but as much as 11% may be owing to cut-off lows which occasionally bring exceptionally heavy rainfall (Abba Omar and Abiodun, 2020).

Cape Town city centre is located between Table Mountain and Table Bay (Fig. 1). Rainfall stations selected for this study are all within 20km of the city centre, and despite receiving variable rainfall amounts, they are influenced by a common rainfall seasonal regime and rain bearing systems. Rainfall records from four weather stations across the Cape Town metropolitan region are investigated; these include Cape Town International Airport, Kirstenbosch, Maitland, and the SAAO (Fig. 1). The SAAO, founded in 1820, is located *c.* 5km east of central Cape Town and *c.* 2km inland of Table Bay, on the

eastern side of Table Mountain. Daily rainfall records began in January 1841 and continue to this day, making it the longest continuous single station daily record in the southern hemisphere (Ndebele et al., 2020). Daily readings have been manually read from a rain gauge located in the gardens of the SAAO over the duration of this period. Given that our aim is to examine long-term (> one century) rainfall changes, for which reliable data become largely unavailable with distance from central Cape Town, and because the nature of rainfall regimes change over relatively small distances (few 10s of km) from this

region, our focus of investigation is constrained to the Cape peninsula only.

### 2.2. Data

As is often the case with early long-term instrumental weather registers, the SAAO record (1841-2018) has some data gaps, owing mainly to the absence of recordings over weekends and public holidays, and in some cases due to instrument failure

or replacement. Although annual rainfall totals are available for all years, daily records are completely absent for the years 1878 and 1879. For the remainder of the record, missing daily rainfall values fall into two categories; namely days recorded as completely absent of data, or as days (e.g. weekends and public holidays) absent of data but with an accumulated total rainfall value on a subsequent day. About 3.2% of missing days have an accumulated value on a subsequent day and only *c.* 2% of the record is completely missing. Imputation of missing values for the SAAO record was performed using data from

neighbouring stations at Maitland (1906-2018), Kirstenbosch (1915-2018), and Cape Town International Airport (1950-2018) (Ndebele et al., 2020). The imputation involves a two-step approach enabling the estimation of wet days and rainfall amounts on those days. One method was applied to days with completely missing records, while a second method was





applied to those with accumulated totals for each month separately as the frequency of wet days and daily rainfall amounts vary with each month (see Ndebele et al., 2020 for further details).


Data for the climate modes: Southern Oscillation Index-SOI (Ropelewski and Jones, 1987), Southern Annular Mode-SAM (Gong and Wang, 1999), and solar (sunspot number) (Vanlommel et al., 2004) were used for examining associations with interannual variability in wet season characteristics. Data are available as either monthly or bimonthly indices for the periods 1876-2018 (SOI), 1851-2011 (SAM), and 1841-2018 (solar).

**2.3.  Defining rainfall variables and season length**

Please refer to Table 1 for terms used to define rainfall variables and the wet season. In this paper, a 'wet day' is defined as one having received $\geq$1mm of rain (as per Camberlin et al., 2009; Singh and Ranade, 2010; Boyard-Micheau et al., 2013; Muthuwatta et al., 2017; Byakatonda et al., 2018). The number of years with complete daily data is represented by $n_d$. The frequency of days with rainfall amounts $> 0mm$ experienced on each calendar day ($d = 1,2,...,365$) over the complete

years is given by $f_{w_1}$. The frequency of wet days ($\geq 1mm$) over the complete years is given by $f_{w_2}$. These frequencies are used to estimate the probability of a wet day ($> 0$mm) $P_{WD}^1$ and ($> 1$mm) $P_{WD}^2$ calculated for each calendar day (Table 1). The average daily rainfall amount is calculated for each calendar day ($\bar{R}_d$), for each month ($\bar{R}_m$), and for each year ($\bar{R}_Y$). The long-term daily average is calculated using all days in the record and is represented by $\bar{R}_L$ (Table 1).

A wet (dry) spell is defined as a period of consecutive days with daily rainfall amounting to $\geq$1mm ($<$1mm) preceded and

followed by a dry (wet) day during the wet season. The term 'one day spell' is used to refer to a wet (or dry) day preceded and followed by a dry (wet) day. On this basis, the following are established: a) frequency of wet/dry spells, b) amount of rainfall during wet spells, c) duration and timing of spells, and d) statistical distributions associated with the lengths of wet/dry spells.

Estimates of wet season onset/termination dates are obtained by applying a combination of methods including the sum of

rainfall values over a five day period (pentads) (after Odekunle, 2004; Amekudzi et al., 2015), and by comparing with the long-term pentad average $\bar{D}_l$ . Initially, data are divided into 73 non overlapping pentads (5 day periods of consecutive days) over 365 days each year, starting on 1 January each year. The 29$^{\text{th}}$ February is not included in affected years and should not adversely impact results as this is a dry time of year. Average rainfall over a 5-day range ($\bar{D}_p$) and long-term pentad average ($\bar{D}_l$) are used to determine the lower and upper limits for wet season onset/termination dates.

For all the stations, days between pentad 19 (starting on Julian day 91) and pentad 61 (ending on Julian day 305), potentially represent the wet season (i.e. 1 April to 31 October). Specific rainfall onset dates ($W_o$) for a calendar year are defined as the first day that the 5-day running sum is above the long-term pentad average ($\bar{D}_l$) occurring after Julian day 90. Rainfall termination dates ($W_E$) are defined as the last day with a 5-day running sum above the long-term pentad average $\bar{D}_l$ occurring before Julian day 306. The wet season ($l_{ws}$) length is then quantified on the basis of onset/termination dates. To





confirm that the period between onset and termination dates actually encapsulates the majority of rainfall received in any given year, the accumulated rainfall during the defined wet season for that year is taken as a proportion of the annual rainfall for that year ($P_{ws}$).

## 2.4. Statistical methods and tests

### 2.4.1. Mann Kendall tests for trends

The Mann Kendall test (Mann, 1945 and Kendall, 1948) is used to test the null hypothesis that there is no trend under the assumption that observations in a series are independent. Given a time series $y_1, y_2, \ldots, y_n$, the following steps were undertaken:

$$S = \sum_{i=1}^{n-1} \sum_{j=i+1}^{n} 1 \times sign(y_j - y_i) \tag{1}$$


where $n$ is the number of data points in the time series $y_1, y_2, \ldots, y_n$; $i = 1, 2, \ldots, n-1$ and $j = 2, 3, \ldots, n$ for $j > i$. The mean of $S$, $E[S] = 0$ and the variance of $S$ is given by:

$$var(S) = \frac{1}{18}[n(n-1)(2n+5)] - \sum_{p=1}^{q} t_p(t_p - 1)(2t_p + 5) \tag{2}$$

where $q$ is the number of tied groups and $t_p$ is the number of data points in the $p$th group.

The test statistic is then calculated as follows:

$$z = \begin{cases} \dfrac{S-1}{\sqrt{var(S)}}, & \text{if } S > 0 \\ 0, & \text{if } S = 0 \\ \dfrac{(S+1)}{\sqrt{var(S)}}, & \text{if } S < 0 \end{cases} \tag{3}$$

The z statistic follows a standard normal distribution and therefore if $|z| > z_{1-\frac{\alpha}{2}}$, the null is rejected and it is concluded that there is a trend present in the time series. Rainfall time series often exhibit serial correlation violating the assumptions of the test, thus adjustments have been made to the Mann Kendall test statistic that allow for dependence in the observations. In this




study, modifications include variance correction (Hamed and Rao, 1998), variance correction using lag-1 correlation (Yue and Wang, 2004), bias corrected pre-whitening (Hamed, 2009) and a bootstrapped optional bias approach (Lacombe et al., 2012). The trend is considered to be significant if at least two of the tests have p-values less than 5% or 10%. Details on these modifications are provided as supplementary information. A multi temporal trend approach is used to trace the change in trend by considering test statistics, p-values and Sen's slopes to measure trend over all possible sub-periods of lengths ranging from 30 years to 100 years.

### 2.4.2. Bivariate wavelet analysis

Bivariate wavelet analysis may be used to test for joint periodicity of two time series. Assuming a continuous time series $y(t)$, a wavelet function is defined as (Joshi et al., 2016):

$$\varphi(\omega) = \varphi(s,\tau) = s^{-\frac{1}{2}}\varphi\left(\frac{(t-\tau)}{s}\right)$$

(4)

where $t$ is time, $\tau$ is the time step in which the window function is iterated, and $s$ is the wavelet scale $(0,\infty)$. The chosen $\varphi(\omega)$ function must have a zero mean and be localised in both time and Fourier space. The CWT is given by the convolution of $y(t)$, scaled and translated to give the function $W(s,\tau)$:

$$W(s,\tau) = s^{-\frac{1}{2}} \int_{-\infty}^{\infty} y(t)\varphi^*\left(\frac{t-\tau}{s}\right) dt$$

(5)

The cross wavelet spectrum of two series $y_1(t)$ and $y_2(t)$ is defined as:

$$W_{y_1,y_2}(s,\tau) = W_{y_1}(s,\tau)W_{y_2}^*(s,\tau)$$

(6)

where $W_{y_1}(s,\tau)$ and $W_{y_2}(s,\tau)$ are the continuous wavelet transforms of $y_1(t)$ and $y_2(t)$ respectively and $*$ denotes the complex conjugate. Two time series may also be analysed for wavelet coherence by the extraction of common periods of high power. This enables the identification of phase relationships between the two series. The wavelet coherence $(R^2)$ of two time series is defined as:

$$R^2(s) = \frac{\left(\left|S\left(s^{-1}W_{y_1,y_2}(s,\tau)\right)\right|^2\right)}{S\left(s^{-1}\left|W_{y_1}(s,\tau)\right|^2\right) \cdot S\left(s^{-1}\left|W_{y_2}(s,\tau)\right|^2\right)}$$

(7)





where *S* is a smoothing operator and is analogous to a localised correlation coefficient in time frequency space. The Morlet wavelet transform is recommended for cross wavelet analysis (Grinsted et al., 2004). Relationships between the wet season variables (wet day frequencies, wet spells, dry spells, season onset and season length) and climate modes including SOI, SAM, and the solar cycles (sunspots) were investigated using bivariate wavelet analysis. The time series were detrended to remove the effect of linear trend and partial wavelet coherence (Ng and Chan, 2012) was calculated where there were correlations between the climate mode indices.

## 3. Results

### 3.1. Overview of dry and wet season characteristics

Average rainfall during the wet season (April to October) ranges from 385mm to 1085mm per annum (Table 2) for the four stations, constituting on average 75% of total annual rainfall. For the common period of 1950-2018, stations on average record 484mm pa at Maitland, 531mm pa at Cape Town International (CPT Int.), 608mm pa at the SAAO and 1422mm pa at Kirstenbosch. Such large spatial rainfall variability is a function of topographic influence. Although the SAAO wet season rainfall record is lower than the average across all stations (av =761mm pa), it is closest to this average value. The number of wet season rain days average 57.4 days for all stations over the period 1950-2018, while the SAAO records on average 55.6 rain days during this period and is thus relatively close to this average. (Table 3). The dry summer months of November to March receive daily rainfall averaging below 0.7mm/day at the SAAO, Maitland and CPT Int., and *c*. 1.5mm/day at Kirstenbosch. Consequently, there are many dry days (<1mm) during the summer months with long dry spells at all stations. In contrast, the cooler seasons (April to October) receive on average *c*. 2mm/day at Maitland and CPT Int., 2.4mm/day at the SAAO and 5.6mm/day at Kirstenbosch, usually peaking in June (Fig. 2).

The number of wet days per annum (January to December), wet days for the wet season (April to October), and the proportion of wet days during the wet season, relative to the total annual number of wet days, are presented for available records in Fig. 3. Over the period 1841-2018, high frequencies of wet days are recorded for the late 19[th] century (SAAO) and during the 1940's and 1950s (all stations) (Table 3). In contrast, the period 1925-1940 records low frequencies of wet days (3 stations). Mann Kendall trend test results indicate a significant decline in the number of wet days during the wet season at the SAAO (by -0.2 days/decade over the period 1841-2018) and at Maitland (by -0.7 days/decade over the period 1906-2018) (Table 3). However, such declines are more pronounced in annual rain day records, which includes the dry season. The frequency of high intensity rainfall (≥ 10mm/day) constitutes on average 5% of days during a calendar year at the SAAO, Maitland and Cape Town Int., and 11.2% of days at Kirstenbosch (see Fig. 3) for the lengths of the records available at each station. Below (above) average frequencies of high intensity rainfall occurred in several years during the 1930s and 1970s (1950s) at all stations since 1900.





A multi temporal trend analysis was performed to identify changes in the linear trends for wet day frequency records at 30-, 50-, 70- and 90-year time scales. The test statistics and Sen's slope at the end of the sub period tested for trend were plotted

for all possible overlapping sub-periods observed for each station record (Fig. 4). At the SAAO, significant positive 30-year trends and 50-year trends are observed for the 1890s and 1900s respectively, mainly owing to the 1880s and early 1890s having had a particularly high frequency of wet days compared to previous decades. In addition, several years between 1926 and 1936 recorded low frequencies compared to the period of high wet day frequencies in the 1940s and 1950s, resulting in significant positive 30-year trends in the latter periods. Outputs from other stations confirm these 30-year trends (Fig. 4).

Negative trends are observed for sub-periods ending in the last 30 years over 50- and 70-year sub-periods, but these are only significant at the SAAO and Maitland.

### 3.2.   Wet and dry spells during the wet season

The average number of wet spells during the wet season ranges between 28 and 31 for all stations over the years 1950-2018,

and average 1.8 days in duration, apart from Kirstenbosch which records a longer average length of 2.3 days (Table 4). There was a prolonged period of below average frequencies of wet spells and average spell lengths at the SAAO and Maitland stations in the 1920s and 1930s, as well as since 1975 for the SAAO (Fig. 5). Average dry spell lengths are between 4.4 and 5.6 days (Fig. 5; Table 4), with maximum spell lengths of between 32 and 48 days. The early 20th century and 1940s had below average dry spell lengths which correspond with periods of higher rainfall (Ndebele et al., 2020). Above average dry

spell lengths between 1925 and 1938 at Maitland are consistent with low rainfall during that period.

A finer-scale assessment of wet/dry spell frequencies at the SAAO is presented in Fig. 6 and Table 5. Wet/dry spell lengths below (and above) the mean spell lengths ($\approx$ 2 days (wet)/ 5days (dry)) are considered and examined over the full study period. The total number of wet spells significantly decreases over the full study period (1841-2018) (Table 5), and most strongly so during the period 1880-1940. One-day events generally have the highest frequencies, averaging 16.5 spells (std

dev = 4.17) over the full study period. The average number of such spells was higher during the period 1941-1957 (av. = 17.2) and lower during the last *c*. 60 years (1958-2018: av. = 15.9), although this becomes higher again during the 21st century (2000-2018; av. = 17.3). There is a significant decline (-1.4 spells/decade) in one-day wet events during the wet season between 1958 and 1999. Two-day wet spells averaged 8.7 spells (std dev = 2.8) per wet season between 1841 and 2018 but decreased substantially over the period 1880-1957 (-0.52 spells/decade). After having increased during the period

1958-1999 (0.87 spells/decade; av. = 9 spells/wet season), two-day wet spells decreased to average 8.5 spells/wet season since 2000. Longer wet spells (> 2 days) averaged 6 spells/wet season between 1841 and 2018, but have progressively decreased in frequency since the 1960s (2000-2018: av. = 4.65 spells/wet season).

Dry spells lasting $\leq$ 5 days (short dry spells) average 21.7 spells per wet season (Fig. 6) and range between 9 spells (1872) and 36 spells (1892). A significant decline (-0.2 spells/ decade) in short dry spells is detected for the full period 1841-2018,

but is most pronounced during the period 1880-1941 (-1.0 spells/decade). Long dry spells (lasting $\geq$ 6 days) average 10 per





wet season (Fig. 6, Table 5), with fewest recorded in 1880 and 1892 (5 spells each), and most in 1930 and 2003 (16 spells each). Long dry spells on average represent 32% of all dry spells during the wet season for the period 1841-2018, but increase to on average 36% since the new millennium, and hence a tendency toward longer dry spells. A significant negative trend (-0.3 spells/decade) is evident prior to 1900, which subsequently changes to a significant increase (0.1 spells/decade)

after 1900. Years with the highest proportion of shorter dry spells ($\leq$ 5 days) include 1883 and 1892 (> 87% of all dry spells), while the highest proportion of dry spells $\geq$ 6 days (40% of all dry spells) occurred in 2010. The period 1880-1920 is notable for its high number of short dry spells (av. = 24.3 spells/yr) and below normal number of longer ($\geq$ 6 days) dry spells (av. = 9 spells/yr) (Fig. 6). In contrast, since *c.* 1974, longer dry spells have for the most part become more frequent (av. since 1974 = 10.7 spells/yr).


### 3.3.  Extreme wet and dry spells during the wet season (SAAO)

The geometric distribution and compound geometric distribution (see Supplementary information) were used to model wet and dry spell lengths, and to estimate the 95[th] and 99[th] percentiles for extreme wet and dry spells. The distribution parameters for the SAAO, Maitland and Cape Town Int. records are similar, and the 95[th] and 99[th] percentiles for wet spell lengths of 4

and 5 days respectively. Several years during the late 19[th] century (1883, 1890, 1892 and 1893) had high frequencies of long wet spells ($\geq$ 4 days) at the SAAO, which coincidently also recorded high rainfall amounts (see Ndebele et al., 2020). The 1940s and early 1950s had the highest recorded number of such extreme wet spells, during which time seven years experienced six or more wet spells of $\geq$ 4 days during the wet season at the SAAO. Long wet spells ($\geq$ 4 days) are also noted during the 1940s and 1950s at Maitland and Cape Town International. The 95[th] percentile for wet spell lengths at

Kirstenbosch = 6 days. The 1940s and 1950s stand out as a period with high frequencies of wet spells $\geq$ 6 days at Kirstenbosch.

The 95[th] and 99[th] percentiles for dry spell lengths are 14 and 20 days respectively (Table 4) at the SAAO. Long dry spells during the months of either April/May or August/September are associated with the onset/termination of wet seasons respectively, and at times reflect a late/early onset/termination to the wet season respectively. Long dry spells (19-28 days in

length) during the mid-wet season (i.e. June/July; Julian days 160-210) were a particular concern in 1929, 1934 and 1935, and coincided with the driest period (1926-1936) since records began in 1841 (see Ndebele et al., 2020). Also noteworthy is the occurrence of long (17 day) dry spells during June/July of recent years (2004, 2005 and 2014).

### 3.4.  Pentad profiles

Decadal pentad profiles for the SAAO are provided so as to obtain a decadal cross-sectional view of the onset, end and distribution of wet season rainfall. Average total values for each pentad over decadal periods (only 9 years for the most recent decade: 2010-2018) were calculated and plotted (Fig. 7). Pentads considered for the wet season are those between





pentad 19 (April 1) and pentad 61 (ending 1 November), as these (and 95% confidence intervals) exceed the long-term pentad average ($\overline{D}_l$) (Fig. 7). Pentads with average values above that of the long-term pentad average are provided in Table 6

and represented in Fig. 8 and Fig. 9. The earliest pentads with mean values above the $\overline{D}_l$ range between pentad 19 (1-5 April) and pentad 30 (26 -30 May) provide an indication of the average wet season onset in each decade. The latest pentads with mean values above $\overline{D}_l$ range between pentad 51 (8-12 August) and pentad 61 (28 October to 1 November). Coincidently, the most recent decade had the latest average wet season onset date, while the first decade had the earliest average wet season termination date. Decades recording the highest number of dry pentads (< 5mm) include 1920-1929 (10 pentads) and 2010-

2018 (13 pentads); these are also the driest decades with peak pentad averages below 20mm. Decades with the most pentads above $\overline{D}_l$  (32-33 pentads), early wet season onsets and late wet season termination dates, correspond with decades that had high rainfall at the end of the 19[th] century and the 1940s and 1950s. The most recent decade (2010-2018) had the least number of pentads above $\overline{D}_l$ (22 pentads). During most decades, rainfall peaks between pentads 28 (1-5 May) and 38 (5-9 July). The peak pentad ranges from an average of 18mm (1930-1939; 2010-2018) to 40mm (1860-1869; 1960-1969). Pentad

profiles for the other stations are provided in the supplementary information. Similar observations are made for the period 2010-2018, which records a tendency toward an increased number of dry pentads, low peak pentad average rainfall, and shortened wet seasons.

### 3.5. Rainfall onset/ termination dates

Rainfall onset/termination dates were calculated for each year across the four weather stations, as shown in Fig. 10 and Table 7. Onset days at the four stations are positively correlated (correlations range between 0.53 and 0.77; Table 8) and have similar averages for the common period 1950-2018, ranging between Julian day 103 (13 April at the SAAO) and 106 (16 April at Cape Town Int. and Kirstenbosch). The latest recorded onset date was at Maitland on Julian day 150 (30 May 1925). Early onsets were common in the 1950s and 1980s (averaging Julian day 100), while prominent late onsets occurred during

the mid-1930s (av. = day 112) and early 1970s (av. = day 117). The average termination dates were Julian day 291/292 (18/19 October) across the four stations, resulting in an average wet season length of 185-188 days. The occurrence of early termination days (before 7 October) has become more prevalent during the most recent 40 years (av = day 284) and constitute 80% of termination dates before Julian day 280 during the last 100 years. Rain season lengths were notably shorter during the period 1970-1994 (av = 181 days) for all stations.

For the SAAO, the average rain season onset and termination dates for the period 1841-2018 are Julian days 103 (13 April) and 291 (18 October) respectively. The latest onset date was 23 May 1933, while the earliest termination date was 26 August 1993. During more recent decades (1958-2018) the mean onset and termination dates have shifted to Julian days 105 (15 April) and 288 (14 October) respectively. Early rainfall onsets (before Julian day 100: 10 April) occurred during the late 1840s to early 1860s, 1940-1960, during the 1980s and 2011-2014. Late rainfall onsets (after Julian day 105; 15 April)

occurred during the decade 1930-1939, the latter part of the 1960s to early 1970s, and many years post 1997. Wet seasons



have typically ended early (on av. before Julian day 289; 17 October) during the most recent decades, starting in 1970. There has been a recent decadal decline in wet season length (-1.4 days/decade) since *c.* 1940, especially between 1941-1994 (-4.2 days/decade), such that wet seasons since 2000 have only averaged 182 days in length while the long-term (1841-2018) average is 188 days.

To test our method of identifying wet seasons, the proportions of annual rainfall accumulated between onset and termination dates were calculated and compared to the totals for January to December ($P_{WS}$). The average proportion of annual rainfall at the SAAO during the identified wet seasons is 81% (1841-2018) (for days ≥ 1 mm). Over the common period 1950- 2018, the other three stations recorded a combined average proportion of 81% of annual rainfall during wet seasons. The most recent four decades (starting 1978) have had the highest proportion (c. 83%) of annual rainfall concentrated in the wet season

at the SAAO. In addition, five years since 1988 have recorded > 90% of annual precipitation during the wet season, which suggests strong inter-seasonal contrasts in rainfall distribution during recent decades.

### 3.6. Associations with climate modes

Continuous bivariate wavelet analysis was used to investigate the relationships between wet season rainfall variables and

possible climate drivers including solar cycles (sunspots), the Southern Annular Mode (SAM), and El Niño Southern Oscillation using the Southern Oscillation Index (SOI). Wet season variables included are wet day frequency, total number of wet spells, short wet spells (< 3 days), long wet spells (> 2days), short dry spells (< 6 days), long dry spells (> 5 days), season onset day and season length. The averages for indices for months between March and October and subsets of these months were used for SAM, SOI and sunspots. Wavelet coherence between wet season variables and climate indices were

analysed in pairs at multiple time scales; these are represented in plots indicating significant coherence at a 5% significance level (Fig. 11). Partial wavelet coherence was considered for SOI and SAM since these indices are correlated.

Figure 11 shows wavelet coherence plots for variables that had significant coherence. Black contour lines in the plots show areas of significance (at 5 % level) of the wavelet coherence. The cone of influence is the blocked area (U shaped) with a white foreground indicating the region where coherence may be inaccurate due to edge effects and zero padding. Significant

relationships are observed between the 9-14 year solar cycle and short (≤ 5 days) dry spells (1908-1964), and also long (≥ 6 days) dry spells (1905-1998). The longer 32-40 year solar cycle significantly correlates with wet season onset days and season length between 1920 and 1975. Although the relationship has persisted into the 2000s, it is not significant after 1975, as this time frame is in the cone of influence. Similarly, at time scales between 32-36 years, there is significant partial coherence between rainfall onset days and the SOI. Wet days, season length and short dry spells show evidence of

significant partial coherence with the SAM over 6-12 year time scales from 1890-1912, 1930-1962 and 1918-1938 respectively. Table 9 provides further details on the dominant time scales for each combination of variables.



## 4. Discussions and conclusions

Previous studies examining rainfall onset/termination dates, numbers of rain days, wet/dry spells and wet season lengths over various parts of southern Africa, have done so using composite datasets but over relatively short periods of investigation (< 60 years) (e.g. Tadross et al., 2005; Tadross et al., 2009; Moeletsi et al., 2011; Philippon et al., 2012; MacKellar et al., 2014). Although our datasets cannot represent a wide geographic region given that it is based on only four station records, these offer a substantially longer-term (up to 178 years) perspective on wet season rainfall characteristics than do records spanning only a few decades. A long record such as we have presented here, is particularly valuable as it can identify trends and interannual variability that may not be detected over shorter (< 60 years) periods of investigation, and in addition, offers an opportunity to test associations between such rainfall characteristics and potential causative mechanisms such as ENSO, SAM, and solar cycles.

While there is a general positive trend in wet day frequencies over Cape Town through the 19[th] century, as reflected in the SAAO data, trends during the 20[th] and early 21[st] centuries are broadly negative. Such trends are similar to those observed by Burls *et al.* (2019) who detected a significant decline in wet days since 1900 using data with more than 100 years from weather stations in the wider western Cape region (including SAAO). We identify two periods with significantly high wet day frequencies; 1881-1905 at the SAAO (av. = 63.6 days) and during the 1940s (av = 66.8 days at 3 stations) and 1950s for all stations (av. = 63.2 days). A decline in wet day frequencies is observed during the common period from 1950 to 2018 for all stations, although this is only significant at Maitland (-1.0 days/decade) and the SAAO (-1.0 days/decade) during the wet season. Declines are also evident over the 103-year period starting in 1915, for three stations (excluding CPT Int.). However, this is only significant for the SAAO during the wet season. The longer SAAO record indicates that for the period 1880-1940, there was a significant decline in wet day frequencies for both the wet season (-0.2 days/decade) and annual series (-0.5 days/decade). The average wet day frequencies during wet seasons of most recent decades (1958-2018, av. = 54.9) is similar to the period 1841-1877 (av. = 56.3) for the SAAO. However, the average frequency of wet days for the full year between these periods has changed more significantly from 74.5 days and 69.1 days respectively, accounting for an overall decrease over the period 1841-2018. The annual number of wet days during summer dry seasons has thus decreased more significantly than during wet seasons over Cape Town. Long-term trends presented here for Cape Town are also consistent with observations made for the more recent period (1960-2010) by MacKellar et al. (2014), who identified declining trends in rain days over large parts of South Africa, including the western Cape region. Studies have also projected declining rain days and an increase in dry spells for future decades (e.g. Abiodun et al., 2017).

Our methodology determined the onset and termination of wet seasons by establishing the first and last days on which the 5-day running sum of rainfall amounts exceed the long-term 5-day average rainfall amounts between April and October each year. This method effectively delineates the wet season [length] as supported by the fact that, on average, it represents 81% of rainfall (≥1mm/day) captured across all years using daily data from the SAAO. Justification for limiting the season onset and end dates to days between 1 April and 31 October is further demonstrated by establishing mean pentad values and their



confidence intervals using available daily data for all four stations. These limits prevent the occurrence of false start/end dates and thus the consequent recording of potential long dry spells at the beginning/end of each wet season. These limits can also be adjusted to suit the local climatology by considering the confidence intervals of mean pentad values. Our method is thus not reliant on the growing season for specific crops, as used by Tadross *et al.* (2005) and Hachigonta *et al.* (2008). This method is reinforced by patterns observed in the pentad profiles across all decades for the period 1841-2018 at the

SAAO. In addition, mid-season dry spells do not affect our defining of the wet season, unlike the method employed by Dunning *et al.* (2016).

Our study demonstrates that wet seasons in Cape Town were longest during the periods 1851-1865, 1890-1913 and 1940-1959 (av. > 195 days), and shortest over the periods 1841-1849, 1930-1939, 1963-1990 and 1998-2010 (av. < 181 days), with most recent trends indicating a shortening of the wet season by 1.4 days/decade since 1941. The identified recent

shortening of the wet season is consistent with findings from other studies for the southwestern Cape region (e.g. Tadross *et al.*, 2009; du Plessis, 2017; Sousa et al., 2018; Ndebele et al., 2020). But shortening wet seasons during recent decades have also been observed for several other sub-Saharan regions of Africa (e.g. Sarr, 2012; Oguntunde et al., 2014; Dunning et al., 2018; Seregina et al., 2019; Sibanda et al., 2020), and is thus a concern for both summer and winter rainfall regions of the continent. Later onset dates for Cape Town are prevalent during the periods 1924-1939, 1961-1980 and 1999-2010, and are

projected to become even further delayed in years to come (Mahlalela et al., 2019). Recent decadal shortening of the Cape peninsular wet season is thus a consequence of both later onset (after Julian day 105; 15 April) and earlier termination (before Julian day 291; 19 October) dates. The pentad profiles provide a decadal overview that combines wet season onset and termination dates, the distribution of rainfall during the wet season, and wet spell lengths. Our data show that some years over the period 2010-2018 had incidences of uncharacteristically high rainfall events during summer months, as also

observed by de Kock et al. (2021). Although the most recent decade (2010-2018) experienced contracted rainfall seasons, the years 2015-2017 mark an outstanding drought, which was not particularly different to the situation over the period 1930-1939. To this end, it is important to place recent drought perspectives in context with both recent decades and longer-term historic events. The implication from the longer record is that past periods of shortened rain seasons with drought were followed again by wetter periods, and that patterns in variability continue into the future given their associations with a

variety of climate modes.

Several previous studies have considered the association between wet season characteristics and climate modes, such as ENSO and the SAM (e.g. Reason et al., 2005; Tadross et al., 2009; Moeletsi et al., 2011; Raut et al., 2014; Mahlalela et al., 2019), but have been temporally constrained in their analyses. Given our 178-year rainfall record, we have now ascertained interannual and interdecadal variability sustained over longer time frames associated with such climate modes. Results

demonstrate that quasi-decadal variability (7-10 years) in onset days and season lengths is associated with the SAM, while variability in dry spell lengths (9-14 years) is connected to the solar cycle. Interdecadal variability ($32-40$ years) in wet season onsets and lengths significantly correlate with long solar cycles of 32-38 years during the 20[th] century, starting from 1920, as also with the SAM. Should such associations still operate to greater or lesser extent during the 21[st] century, then





some of the recent late onset dates and shorter seasons might be attributed to the effects of SOI and SAM variability. Such
associations suggest that these climate modes, in combination, have influenced seasonal characteristics of rainfall over Cape
Town and adjacent regions during the recent past, and are likely to continue to exercise an influence in future (to a greater or
lesser extent). Discussions concerning recent, current and future (forthcoming years) wet season characteristics should thus
take cognisance of the underlying interannual variability associated with these climate modes. This has important
implications for appropriate forecasting and planning of forthcoming wet seasons in the region.


**Acknowlegdements**

We thank the South African Weather Service for providing rainfall data.

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





**Tables**

Table 1: Definitions of terms

| Term | Symbol | Calculation |
|---|---|---|
| Proportion of wet days > 0mm (per Calendar day) | $P_{WD}^1$ | $$P_{WD}^1 = \frac{f_{w_1}}{n_d}$$ |
| Proportion wet days $\geq$ 1mm (per Calendar day) | $P_{WD}^2$ | $$P_{WD}^2 = \frac{f_{w_2}}{n_d}$$ |
| Average daily rainfall (per Calendar day) | $\overline{R}_d$ | $$\overline{R}_d = \frac{\sum_{i=1}^{n_d} R_d^i}{n_d} \text{ for } d = 1,2,\dots,365$$ |
| Average daily rainfall (per month) | $\overline{R}_m$ | $$\overline{R}_m = \frac{\sum_{i=1}^{n_m} R_m^i}{n_m} \text{ for } m = 1,2,\dots,12$$ |
| Average daily rainfall (per year ) | $\overline{R}_Y$ | $$\overline{R}_Y^i = \frac{\sum_{d=1}^{365} R_d^i}{365} \text{ for } i = 1,2,\dots,n_d$$ |
| Long term daily average (full record) | $\overline{R}_L$ | $$\overline{R}_L = \frac{\sum_{i=1}^{n_d} \overline{R}_Y^i}{n_d}$$ |
| Pentad rainfall totals | $D_P$ | $$D_P^i = \sum_{d=5(p-1)+1}^{5p} R_d^i \quad \begin{array}{l} \text{for } p = 1,2,\dots 73; \\ i = 1,2,\dots,n_d \end{array}$$ |
| Pentad averages | $\overline{D}_p$ | $$\overline{D}_p = \frac{\sum_{i=1}^{n_d} D_p^i}{5n_d} \text{ for } p = 1,2,\dots,73$$ |
| Long term pentad average | $\overline{D}_l$ | $$\overline{D}_l = \frac{\sum_{p=1}^{73} \overline{D}_p}{73}$$ |
| Wet season length | $l_{ws}^i$ | $l_{ws}^i = W_E - W_o$ |
| Proportion of wet season rainfall | $P_{ws}$ | $$P_{ws} = \frac{\sum_{d=ws} R_d}{\sum_{k=1}^{12} R_m}$$ |



Table 2: Information on rainfall stations. The mean total annual rainfall for the rainfall series and mean seasonal rainfall (Apr –Oct) is presented for each station. R = correlation between the Observatory daily rainfall record and that for other stations over common recording periods.


| Name | Location | Years | Length | Annual | Apr-Oct | R |
|---|---|---|---|---|---|---|
| Observatory | -33.9330; 18.4770 | 1841 – 2018 | 178 | 619.19 | 483.6 | 1 |
| Maitland | -33.9200; 18.5060 | 1906 – 2018 | 112 | 493.96 | 385.3 | 0.70 |
| Kirstenbosch | -33.9830; 18.4330 | 1915 – 2018 | 104 | 1384.63 | 1184 | 0.63 |
| Cape Town Int. | -33.9670; 18.6000 | 1950 – 2018 | 69 | 531.33 | 414.4 | 0.74 |





Table 3: Wet day frequency (days) for various stations over the Cape Town Metropol. Significant (Sig) trends have Mann

Kendall (MK) test p-values < 0.10 (at 10% significance level) and p-values < 0.05 (at 5% significance level) for at least 2 of

the modified MK tests.

| Station (years) | Months | Mean (std. dev) | Min (year) | Max (year) | Sen's slope | MK p-value | Sig. (Y/N) |
|---|---|---|---|---|---|---|---|
| SAAO (1841-2018) | April to October | 57.85 (8.9) | 39 (1880) | 83 (1892) | -0.2 | < 0.10 | Y |
| | Annual | 74.59 (10.5) | 51 (1880) | 101 (1883) | -0.5 | < 0.05 | Y |
| Maitland (1906-2018) | April to October | 50 (9.9) | 31 (1936) | 69 (1957) | -0.7 | < 0.05 | Y |
| | Annual | 62.1 (8.3) | 42 (1931) | 87 (1913) | -1.0 | < 0.05 | Y |
| 1915-2018 | | | | | | | |
| SAAO | April to October | 56.9 (8.8) | 40 (1927) | 80 (1941) | -0.74 | < 0.05 | Y |
| | Annual | 72.7 (10.4) | 52 (2000) | 100 (1941) | -1.25 | < 0.05 | Y |
| Kirstenbosch | April to October | 71.4 (8.7) | 56 (1985) | 91 (1941) | -0.30 | > 0.10 | N |
| | Annual | 92.5 (10.1) | 72 (1979) | 117 (1941) | -0.40 | > 0.10 | N |
| Maitland | April to October | 49.5 (8.3) | 31 (1936) | 69 (1957) | -0.60 | > 0.10 | N |
| | Annual | 61.2 (9.7) | 42 (1931) | 86 (1957) | -0.72 | < 0.10 | Y |
| 1950-2018 | | | | | | | |
| SAAO | April to October | 55.6 (8.3) | 42 (1963) | 75 (1951) | -1.010 | < 0.10 | Y |
| | Annual | 70.5 (9.8) | 52 (2000) | 94 (1951) | -1.43 | < 0.05 | Y |
| Kirstenbosch | April to October | 71.0 (8.2) | 56 (1985) | 91 (1970) | -0.83 | > 0.10 | N |
| | Annual | 92.0 (9.7) | 72 (1979) | 116 (1970) | -0.87 | > 0.10 | N |
| Maitland | April to October | 48.9 (7.5) | 35 (2015) | 69 (1957) | -1.11 | < 0.05 | Y |
| | Annual | 60.5 (9.4) | 42(2000) | 86 (1957) | -1.23 | < 0.05 | Y |





| Cape Town Int. | April to October | 54.1 (8.8) | 39 (2015) | 76 (1951) | 0.00 | > 0.10 | N |
| | Annual | 67.9 (7.9) | 49 (1999) | 88 (1951) | -0.30 | > 0.10 | N |



Table 4: Summary statistics of wet and dry spell lengths (≥ 1mm/ day) and average total spells (ATS) per wet season for

1950-2018 at all stations. The 95 percentile = 95[th] pct and 99[th] percentile =99[th] pct.

|  | Station | Max | Mean | Median | Std Dev | 95[th] pct | 99[th] pct | ATS |
|---|---|---|---|---|---|---|---|---|
| Wet spells | SAAO | 14 | 1.8 | 1 | 1.14 | 4 | 5 | 31 |
|  | CPT Int. | 11 | 1.8 | 1 | 1.22 | 4 | 5 | 30 |
|  | Fire station | 9 | 1.8 | 1 | 1.21 | 4 | 6 | 29 |
|  | Kirstenbosch | 15 | 2.3 | 2 | 1.58 | 6 | 8 | 32 |
|  | Maitland | 9 | 1.8 | 1 | 1.13 | 4 | 5 | 29 |
| Dry spells | SAAO | 42 | 5 | 4 | 4.46 | 14 | 20 | 31 |
|  | CPT Int. | 35 | 5.3 | 4 | 4.53 | 14 | 21 | 30 |
|  | Fire station | 36 | 4.8 | 3 | 4.47 | 11 | 21 | 29 |
|  | Kirstenbosch | 32 | 4.4 | 3 | 3.84 | 12 | 17 | 32 |
|  | Maitland | 48 | 5.6 | 4 | 5.32 | 15 | 23 | 29 |





Table 5: Summary statistics of wet and dry spell lengths during the wet season at the SAAO (1841-2018). Significant (Sig)

trends have Mann Kendall (MK) test p-values < 0.10 (at 10% significance level) and p-values < 0.05 (at 5% significance

level) for at least 2 of the modified MK tests.

| | Spell lengths | Period (years) | Average | Sen's slope spells/decade | MK p-value | Sig Y/N |
|---|---|---|---|---|---|---|
| Wet spells | 1-14 days | 1841-2018 | 31 | -0.17 | < 0.05 | Y |
| | 1-14 days | 1880 - 1940 | 28 | -0.83 | < 0.05 | Y |
| | 1 day | 1841-2018 | 16.5 | -0.08 | > 0.10 | N |
| | | 1841-1940 | 16.8 | 0.00 | > 0.10 | N |
| | | 1941-1957 | 17.2 | 1.60 | > 0.10 | N |
| | | 1958-2018 | 15.9 | 0.00 | > 0.10 | N |
| | | 2000-2018 | 17.3 | -1.40 | > 0.10 | N |
| | 2 days | 1841 – 2018 | 8.7 | 0.00 | > 0.10 | N |
| | | 1890-1922 | 9.7 | 0.00 | > 0.10 | N |
| | | 1923-1957 | 7.3 | 0.00 | > 0.10 | N |
| | | 1958-1999 | 9 | 0.77 | > 0.10 | N |
| | | 2000-2018 | 8.5 | 0.00 | > 0.10 | N |
| | 3 – 14 days | 1841-2018 | 6 | 0.00 | > 0.10 | N |
| | | 1940-1957 | 7.8 | 0.00 | > 0.10 | N |
| | | 1958-2018 | 5.3 | 0.00 | > 0.10 | N |
| Dry Spells | ≤ 5 days | 1841 - 2018 | 21.7 | -0.20 | < 0.05 | Y |
| | | 1880-1940 | 19.7 | -1.00 | < 0.05 | Y |
| | > 5 days | 1841-2018 | 10 | 0.00 | > 0.10 | N |





| | | 1841-1899 | 9.5 | -0.37 | < 0.05 | Y |
|---|---|---|---|---|---|---|
| | | 1900-2018 | 10.4 | 0.00 | < 0.10 | Y |




Table 6: Pentad profiles per decade at the SAAO. Pentad long-term mean = 8.44mm and dry pentads have < 5mm of rain.
Pentad 19-24 (1-30 April); Pentad 25-30 (1-30 May); Pentad 51-55 (8 September -2 October); and Pentad 56-61(3 October –
1 November).

| Decade | Lowest pentad above mean | Highest pentad above mean | N° of Pentads above mean | N° of dry pentads | Peak pentad | Peak Pentad rainfall |
|---|---|---|---|---|---|---|
| 1841 – 1849 | 20 | 51 | 27 | 7 | 32 | 27.7 |
| 1850 – 1859 | 19 | 57 | 29 | 6 | 28 | 33.8 |
| 1860 – 1869 | 23 | 61 | 30 | 1 | 36 | 35.5 |
| 1870 – 1877 | 22 | 57 | 29 | 4 | 31 | 28.4 |
| 1880 – 1889 | 21 | 57 | 31 | 2 | 34 | 25.2 |
| 1890 – 1899 | 23 | 61 | 32 | 4 | 43 | 30 |
| 1900 – 1909 | 19 | 60 | 30 | 3 | 33 | 29.8 |
| 1910 – 1919 | 19 | 55 | 30 | 8 | 39 | 29.5 |
| 1920 – 1929 | 24 | 56 | 25 | 10 | 35 | 25.8 |
| 1930 – 1939 | 20 | 56 | 25 | 8 | 44 | 18.8 |
| 1940 – 1949 | 19 | 61 | 32 | 7 | 30 | 33.3 |
| 1950 – 1959 | 20 | 60 | 32 | 3 | 28 | 30.1 |
| 1960 – 1969 | 21 | 58 | 23 | 8 | 32 | 35.4 |
| 1970 – 1979 | 24 | 59 | 27 | 6 | 33 | 27.7 |
| 1980 – 1989 | 20 | 56 | 30 | 8 | 48 | 32.6 |
| 1990 – 1999 | 20 | 59 | 30 | 6 | 35 | 27.3 |
| 2000 – 2009 | 20 | 56 | 33 | 6 | 38 | 31.9 |
| 2010 – 2018 | 30 | 61 | 22 | 13 | 42 | 17.4 |





Table 7: Seasonal summary statistics for onset, termination and length of the wet season.

| Station (years) | Average Onset | Latest Onset | Average End | Earliest End | Average Length | Min Length | Max Length |
|---|---|---|---|---|---|---|---|
| SAAO (1841-2018) | 103 | 143 | 291 | 238 | 188 | 135 | 214 |
| Maitland (1906-2018) | 105 | 150 | 292 | 253 | 187 | 149 | 214 |
| 1915-2018 | | | | | | | |
| SAAO | 104 | 143 | 290 | 238 | 186 | 142 | 214 |
| Kirstenbosch | 106 | 140 | 291 | 240 | 185 | 141 | 214 |
| Maitland | 105 | 150 | 291 | 253 | 186 | 149 | 214 |
| 1950-2018 | | | | | | | |
| SAAO | 103 | 143 | 291 | 238 | 188 | 135 | 214 |
| CPT Int. | 106 | 145 | 291 | 226 | 186 | 127 | 213 |
| Kirstenbosch | 106 | 140 | 291 | 240 | 185 | 141 | 214 |
| Maitland | 105 | 150 | 292 | 253 | 187 | 149 | 214 |




Table 8: Wet season characteristic correlations for the four stations: 1950-2018. O = onset day, T = termination day, L= wet season length. 1- Observatory, 2- Maitland, 3 – Kirstenbosch and 4 – Cape Town International. Shaded correlations are for similar characteristics.

|     | O1 | T1 | L1 | O2 | T2 | L2 | O3 | T3 | L3 | O4 | T4 | L4 |
|-----|-----|-----|-----|-----|-----|-----|-----|-----|-----|-----|-----|-----|
| O1 | 1.000 | | | | | | | | | | | |
| T1 | 0.019 | 1.000 | | | | | | | | | | |
| L1 | -0.616 | 0.776 | 1.000 | | | | | | | | | |
| O2 | 0.532 | 0.058 | -0.289 | 1.000 | | | | | | | | |
| T2 | 0.045 | 0.445 | 0.322 | 0.062 | 1.000 | | | | | | | |
| L2 | -0.326 | 0.306 | 0.447 | -0.631 | 0.735 | 1.000 | | | | | | |
| O3 | 0.647 | -0.021 | -0.425 | 0.603 | -0.185 | -0.553 | 1.000 | | | | | |
| T3 | 0.072 | 0.526 | 0.369 | 0.141 | 0.539 | 0.323 | -0.017 | 1.000 | | | | |
| L3 | -0.380 | 0.399 | 0.554 | -0.300 | 0.519 | 0.607 | -0.682 | 0.743 | 1.000 | | | |
| O4 | 0.685 | 0.016 | -0.419 | 0.578 | -0.075 | -0.451 | 0.765 | 0.002 | -0.511 | 1.000 | | |
| T4 | 0.003 | 0.687 | 0.540 | 0.013 | 0.432 | 0.327 | -0.093 | 0.681 | 0.560 | -0.105 | 1.000 | |
| L4 | -0.439 | 0.470 | 0.648 | -0.363 | 0.351 | 0.519 | -0.558 | 0.475 | 0.721 | -0.718 | 0.768 | 1.000 |






Table 9: Wavelet coherence between wet season characteristics and climate mode indices (CMI): Sunspots (SS), Southern Oscillation Index (SOI), Southern Annular Mode (SAM) and Sea Surface Temperatures (SST). Partial wavelet coherence between the wet season characteristic and CMI[1] by removing the effect of CMI[2] where it has been included. All variables are considered of the months indicated where February – March (FM), March-May (MAM) and April – October (WS). Periods that are significant (Sig) at a 5% level of significance are given with corresponding Calendar years estimates.

| Wet season characteristic | CMI[1] | CMI[2] | Months | Sig. periods (years) | In phase (IP) /out of phase (OP) | Sig. Calendar Years |
|---|---|---|---|---|---|---|
| Onset day | SS | | MAM | 32-35 | OP | 1920-1975 |
| Season length | SS | | WS | 32-35 | IP | 1909-1973 |
| | SAM | SOI | WS | 6-11 | OP | 1927-1959 |
| Wet days | SAM | SOI | JJA | 6-8 | OP | 1890-1912 |
| Wet spells (> 2 days) | SS | | WS | 32-38 | OP | 1895-1973 |
| Dry spells (≤ 5 days) | SS | | WS | 9-16 | OP | 1905-1962 |
| | SAM | SOI | WS | 8-12 | OP | 1920-1940 |





**Figures**


Figure 1: Location of the SAAO and other stations in Cape Town, southwestern Cape of South Africa (Ndebele et al., 2020).





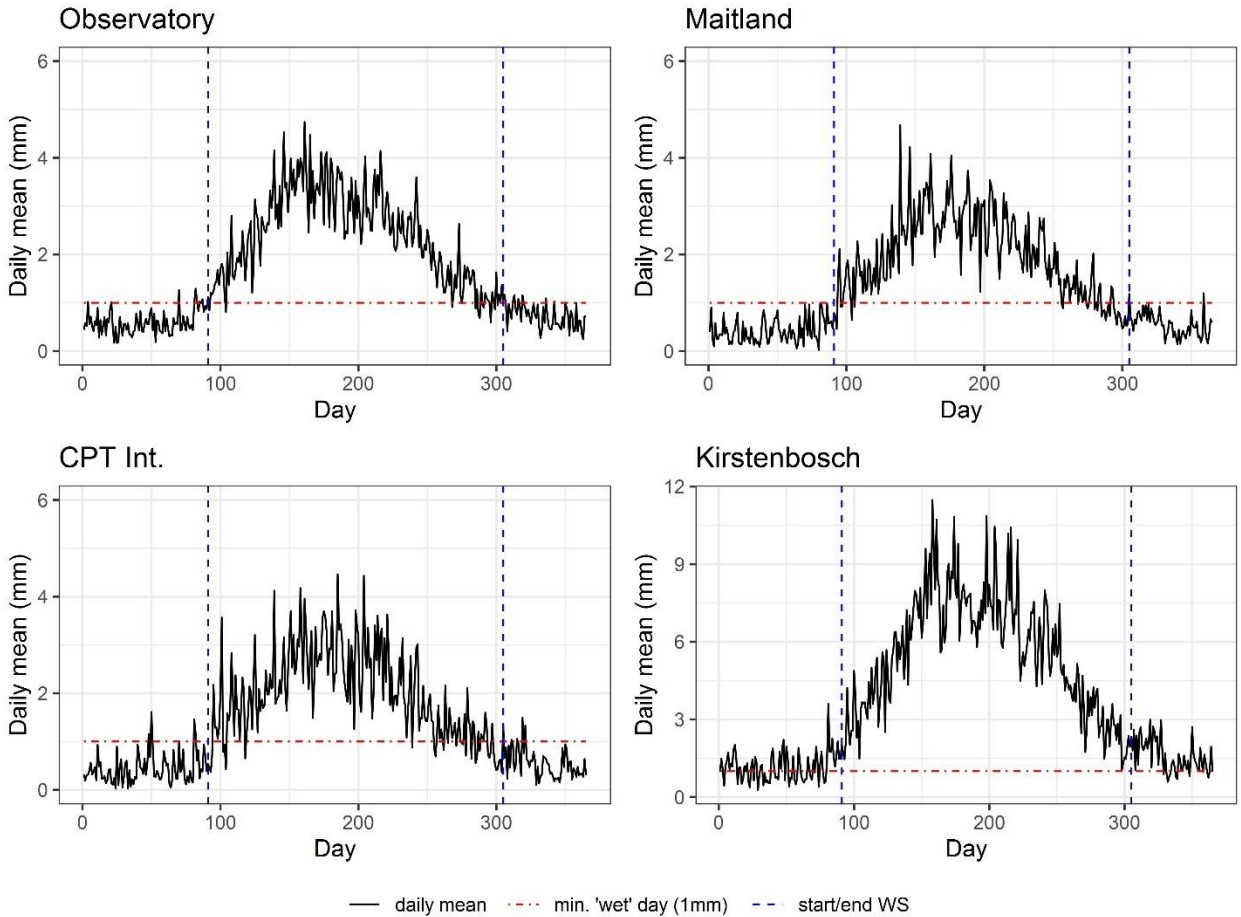

Figure 2: Average daily mean for each Julian day calculated over the full record available at each station – Observatory (1841-2018), Maitland (1906-2018), Kirstenbosch (1915-2018) and Cape Town Int. (1950-2018). Day 91 (1 April) and day 305 (31 October) represented the start and end of the wet season (WS) respectively. The minimum for wet day (min. wet day) is 1mm.

**Climate**
**of the Past**
Discussions




Figure 3: The annual and wet season (April to October) frequency of wet days (≥ 1mm) and the corresponding proportion relative to the total number of days (365) per year in the second vertical axis.




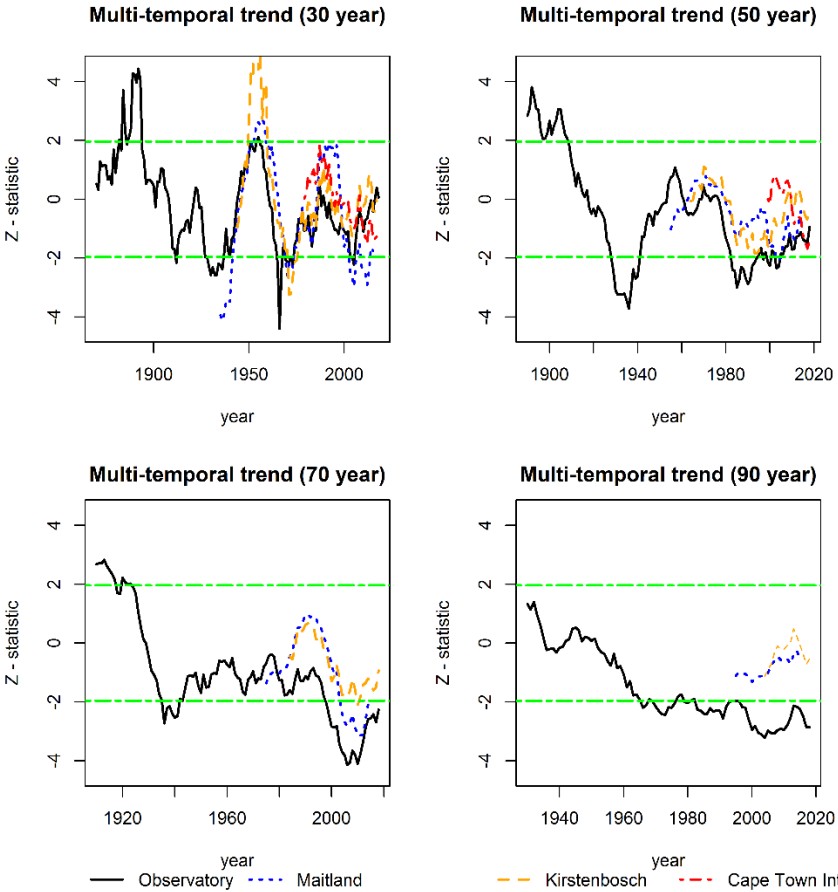

Figure 4: The test statistic values for the multi-temporal trends calculated over overlapping 30-, 50-, 70- and 90-year sub-periods for the 4 stations. The end year for each sub-period is given as the horizontal axis and the z test statistic for the modified Mann Kendall trend test applied is plotted against the last year included in the sub-period. The statistic $z = -1.96$ and $z = 1.96$ corresponds with the 5% level of significance (red horizontal dashed lines). Test statistic values above (below) $z = 1.96$ ($z = -1.96$) represent significant trends.





Figure 5: Total number of wet and dry spells per wet season (frequency) and mean spell lengths (MSL) for wet and dry spells during the wet season.



Figure 6: Wet/dry spell frequencies at the SAAO.





Figure 7: Mean pentad rainfall (MR) and 95% confidence intervals. Horizontal dashed blue lines indicate threshold for wet days (5mm) and long-term mean of all pentads respectively. Vertical dashed blue lines the start (pentad 19) and end (pentad 61) of the wet season.





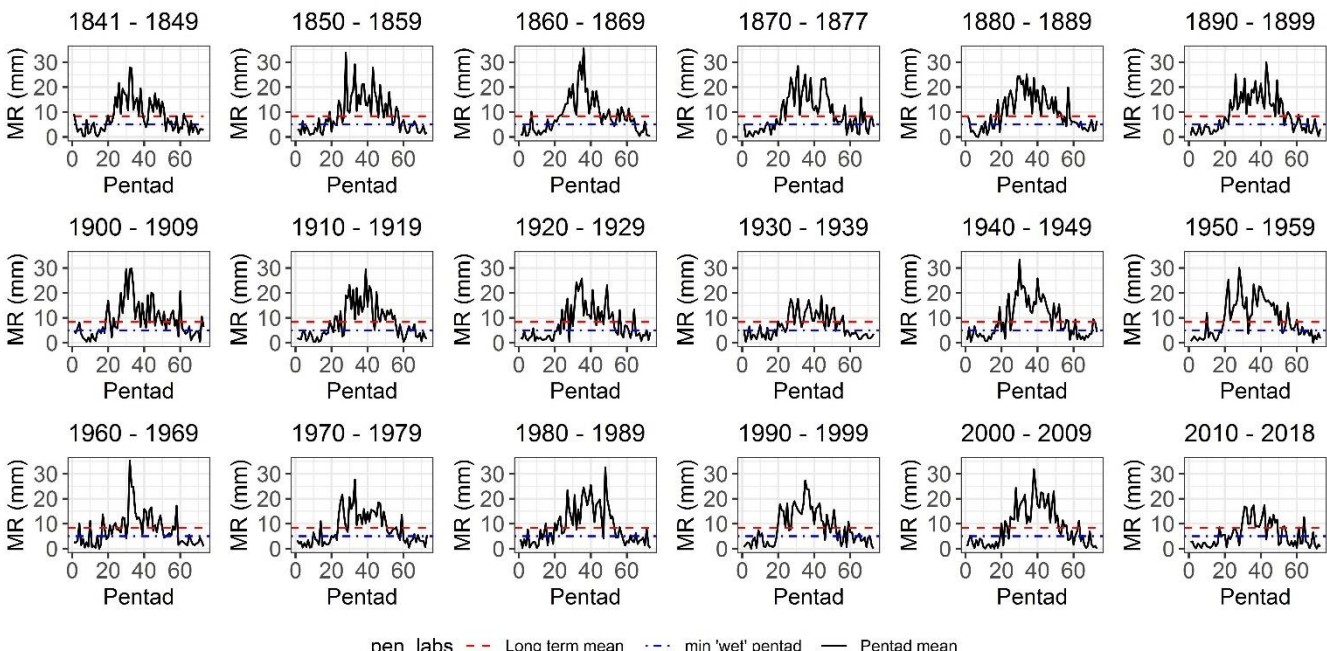

Figure 8: Pentad rainfall profiles: mean rainfall (MR) pentad totals for each decade.





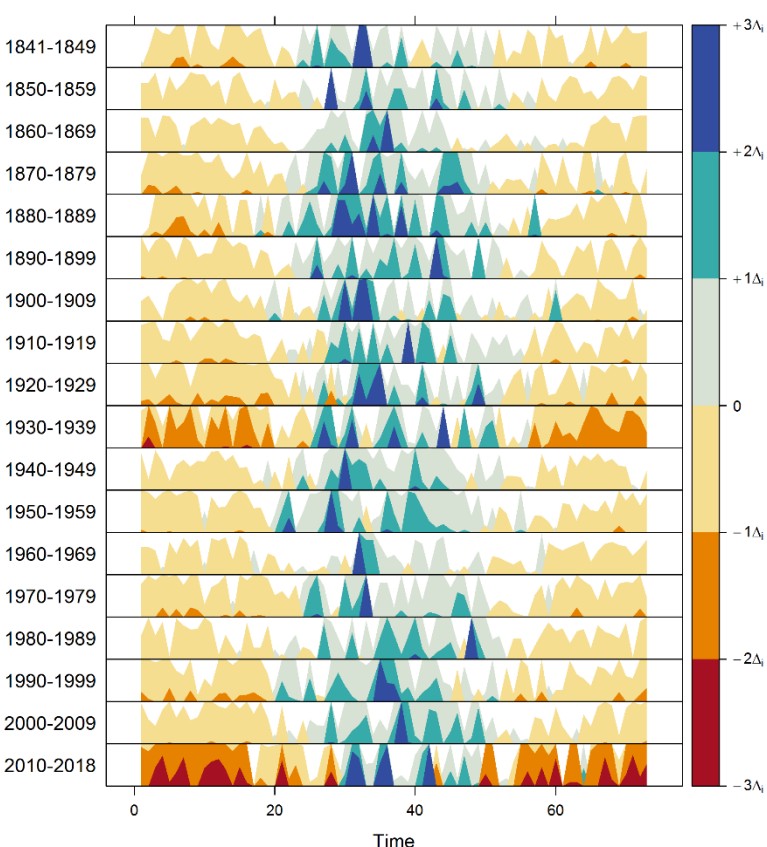

685

Figure 9: The horizon plots are plotted for difference in the pentad mean relative to the overall long term pentad mean for the
series. Positive values (blue) represent pentad mean values above the overall long term pentad mean and negative values
690                                    indicate pentad mean values below the overall long term mean.



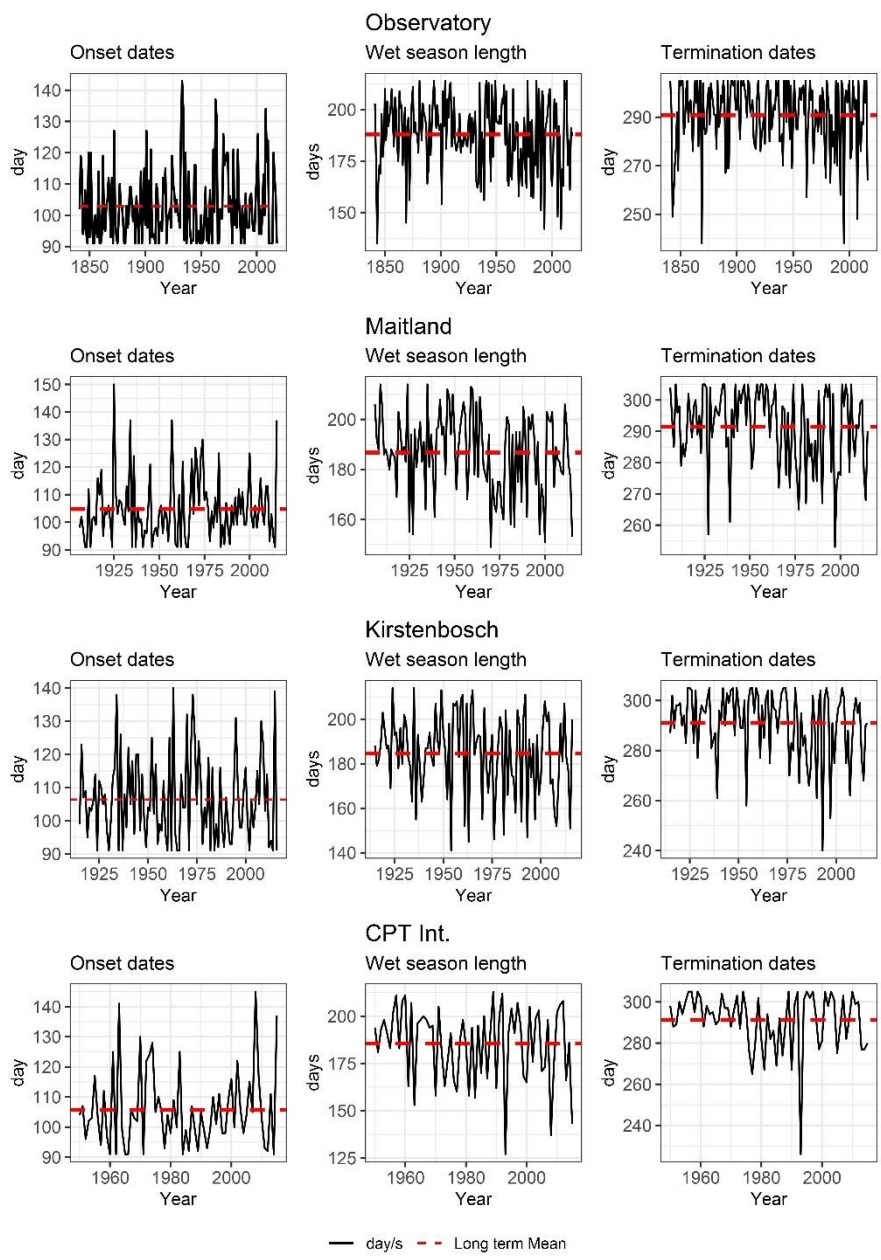

Figure 10: Onset/termination and length of wet seasons during the wet season.

695



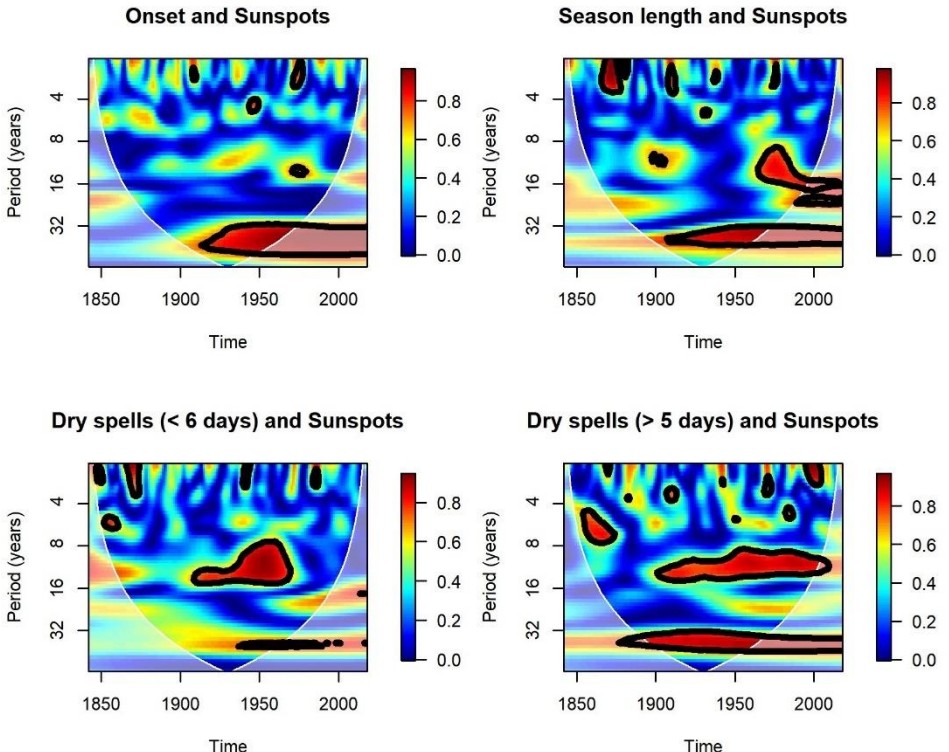

Figure 11: Wavelet coherence plots: seasonal rainfall characteristics with solar [sunspots]. The scale for shaded colours in the plot represents wavelet coherence and is scaled from lowest (blue) to highest (red). Black contour lines show areas of significance (at 5 % level) of the wavelet coherence. The cone of influence is the blocked area (U shaped) with a white foreground.

700