# Peer review of "Wet season rainfall characteristics and temporal changes for Cape Town South Africa, 1841-2018"

_Climate of the Past, 2021_

## Author Response (AR1)

**Reviewer 1 Comments and responses**:

My two main comments are that the authors need to justify the significance of this work more powerfully, why is this important and to whom, and demonstrate the quality of the data at this station early on, recognising human activities and management of the station are also important.

There is little assessment of the data quality at the site, you are using a long series, have there been changes in instrument, rainfall recording practice, location, even when to human recorders change – these are all likely/certain to varying degrees, but are important points to consider and can help explain potential changes in the data. This might explain why there was a change in pentads in the 1930s and 2010s, or it might be climatic variability. Irrespective of cause, demonstrating this understanding will strengthen your arguments and conclusions (easy to add around line 140).

The significance of the paper is commented upon by the authors, but my key point on completing the paper was it fails to demonstrate the need for the study – the 'so what' question. The paper would be much stronger if you could demonstrate why a ~5-day shorter rainy season is important, what impact will this have? This should be quite easy to add, but demonstrate it rather than just stating it will have an impact on water management...

I would encourage you to separate the discussion and conclusions – this will permit you to discuss the findings within the context of the wider literature and then highlight and reiterate the key points from this study.

I think you can reduce the number of tables and figures presented, some appear to offer limited additional information on that already presented within the text (comments below).

**Response**:

We have addressed the data quality question and expanded on this, mentioning instrumental changes etc – please see lines 130 – 134. We also refer the reader to Ndebele et al. (2020) who discuss the data quality and cleaning process in more detail.

We have separated the discussion and conclusion as requested. The conclusion details the practical significance of our work.

| No. | Reviewer 1 Minor comments | Response |
|-----|---------------------------|----------|
| 1 | Line ~135  Do you get any hail/snowfall? It might be worth adding a sentence stating as a justification for the use of rainfall rather than precipitation. | The records do not indicate whether there was hail or snowfall. However, snowfall is unknown at the recording stations and hail is a rare occurrence (less than one event per annum) |

| 2 | Line ~140 What about trace precipitation measurements >0mm but <1mm. Please clarify. | A day with rainfall < 1mm is considered a dry day. Wet and dry spells are also defined using this threshold. |
|---|---|---|
| 3 | Line 171 add space between '1and' | Line 185
Corrected to "1 and" |
| 4 | Line 219 remove ' from 1940's, so 1940s | Line 235
Removed ' and changed to 1940s |
| 5 | Line 226 end sentence after …(1950s). delete at all stations since 1900. | Line 242
Ended the sentence at 1950s |
| 6 | Line~320-23 is this shift in dates significant or just noise? | It is not significant. |
| 7 | Tables – are all these needed, I think there may be an opportunity to reduce the number presented. Reconsider Tables 2, 4, 5, 7 & 9, particularly Tables 2 & 7 – do these add anything not within the text? | Table 4, 5, 7 and 9 have been removed and including in supporting documents. |
| 8 | Figure 3 – Difficult to see red line (A-O 5-year Gaussian filter) | Line 664
Changed the red line so that it is thicker and more visible. |
| 9 | Figure 5 MSL – is this days? | Line 676
Added 'in days' |
| 10 | Figure 6 – would these benefit from a 10 or 30 year running mean? I ask as looking at >3 days there looks to be an underlying pattern that is deviated from in ~1870-1910 and ~1940-1960. | Line 680
Added a 10 year Gaussian filter to the plot. |
| 11 | Figure 8 – remove? | This figure was kept in the manuscript. |
| 12 | Figure 9 – I think this is a powerful graphic, but the caption could be revised to be more explicit and help the reader see more clearly what is being | Line 695
The x-axis label was changed to 'Pentads'.
Line 699 – 702
Added further explanation to the caption. |

| | | |
|---|---|---|
| presented. Revise x-axis label – time units? | | |

**Reviewer 2 Comments and responses**

**Major comment**

The main concern I have with this paper is the conclusions drawn from the complex wavelet analysis, particularly in relation to the role of solar variability on rainfall in southern Africa. My understanding is that the impact of solar variation on regional rainfall is likely to be very small, and that modern studies have found a correlation, but no real causation. At the moment these results seem to be the product of statistics, without any connection to what is happening on the ground. If that is the goal of the study, then that needs to be made clearer, but I think consideration of the dynamics would make the paper much more convincing.

The easiest way to address this is to provide additional information in the introduction and conclusion about how solar variations, ENSO and SAM dynamically influence the weather and climate of Cape Town. Perhaps it is worth summarising the key results from the other studies mentioned, for example.

**Response:**

We now provide additional information from referenced sources (published in high ranking international and peer reviewed journals) which show good correlation between solar forcing and rainfall in the region. Although our results agree with and expand upon such past study outcomes, we do not place a strong emphasis on solar forcing, other than to show the association, say that there is a relationship,and that it should be one that deserves consideration. We have highlighted sections (please see lines 71 to 83).

| | Reviewer Comment 2 Minor Comments | Response |
|---|---|---|
| 1 | Lines 17–18: Is a decline of 3 days statistically significant? If so, it should be mentioned. | Lines 17 – 19 These lines have been reworked accordingly. |
| 2 | Bottom of page 2: Could a gauge reading of 0.1mm also indicate dew, rather than rainfall? | Although this might be possible it is likely to be rare. The records |

| | | do not specify the type of precipitation. |
|---|---|---|
| 3 | Line 98: 'and also some' rather than 'as also some' | Line 108 changed to 'and also some' |
| 4 | Line 130–135 could be expanded a little, with more detail added. Perhaps a table can be included to provide more specific detail about the climate mode indices used, their frequency, and the exact dataset used for their derivation. Which dataset was used to extend the Gong and Wang SAM index back to 1851, for example? Presumably 20CR, but it would be good to clarify this, particularly because there may be some quality issues examining SAM that far back | Line 616 A table (Table 1) has been added with details of the data sets. |
| 5 | Line 189: Can you please spell out CWT? | Line 205 Changed CWT to continuous wavelet transform |
| 6 | Line 318-319: Are the lengths significantly shorter as well? It would be good to clarify this. | The lengths are not significantly shorter but if they continue in a similar trend will become significant. |
| 7 | Line 326: Is 17 October Julian day 290, not 289? | Line 342 Yes this has been changed from 289 to 290. |
| 8 | Line 329-330: This is a dramatic statistic that might go better in the abstract than the current information provided in lines 17–18. | Line 17-18 in the abstract have been removed and changed to include these statistics – mean season length since 2000 is 182 days while the overall long term mean is 188. |
| 9 | Figure 3: Is it possible to replot these graphs to be longer, with the same x-axis and stacked on top of each other as four long plots rather as a 2x2 of square plots? I think this would allow for easier comparison across the stations, and make it easier to see the interannual variability. | Line 664 This figure has been changed such that the plots for the four stations are stacked over each other and have the same x-axis and y axis. |
| 10 | Figure 8: Presumably this figure is for SAOO? | Line 690 Added 'at SAAO' in the caption. |